# TEXT-TO-3D GENERATION WITH BIDIRECTIONAL DIFFUSION USING BOTH 2D AND 3D PRIORS

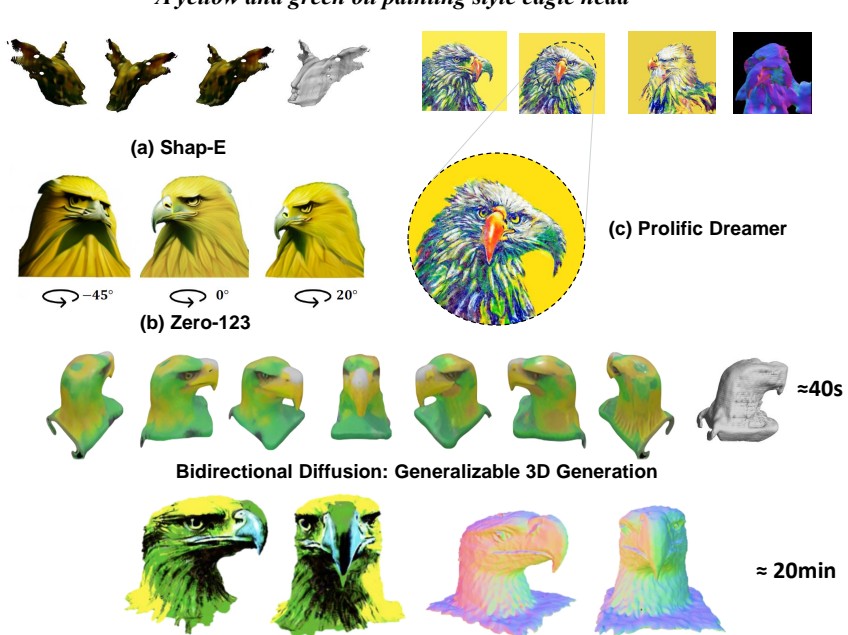

"A yellow and green oil painting style eagle head"

(a) Shap-E

(b) Zero-123

(c) Prolific Dreamer

Bidirectional Diffusion: Generalizable 3D Generation

≈40s

Improve the Geometry and Efficiency of Optimization-based Methods

≈ 20min

Figure 1: Our *BiDiff* can efficiently generate high-quality 3D objects. It alleviates all these issues in previous 3D generative models: (a) low-texture quality, (b) multi-view inconsistency, and (c) geometric incorrectness (e.g., multi-face Janus problem). The outputs of our model can be further combined with optimization-based methods (e.g., ProlificDreamer) to generate better 3D geometries with slightly longer processing time (bottom row).

## ABSTRACT

Most research in 3D object generation focuses on boosting 2D foundational models into the 3D space, either by minimizing 2D SDS loss or fine-tuning on multi-view datasets. Without explicit 3D priors, these methods often lead to geometric anomalies and multi-view inconsistency. Recently, researchers have attempted to improve the genuineness of 3D objects by training directly on 3D datasets, albeit at the cost of low-quality texture generation due to the limited 2D texture variation in 3D datasets. To harness the advantages of both approaches, in this paper, we propose *Bidirectional Diffusion* (*BiDiff*), a unified framework that incorporates both a 3D and a 2D diffusion process, to preserve both 3D fidelity and 2D texture richness, respectively. Recognizing that a simple combination can yield inconsistent generation results, we further bridge them with innovative bidirectional guidance. Moreover, we further offer an optional refinement phase utilizing the denoised results to initialize the optimization-based methods, markedly addressing the geometry incorrectness problem and improving the efficiency(3.4h → 20min). Experimental results have shown that our model achieves high-quality, diverse, and scalable 3D generation. The project website is https://bidiff.github.io/.

Figure 2: Sampling results of Bidirectional Diffusion model. *BiDiff* can separately control texture generation (a) and geometry generation (b).

(a) **Texture Control**: we change the texture while maintaining the overall shape.

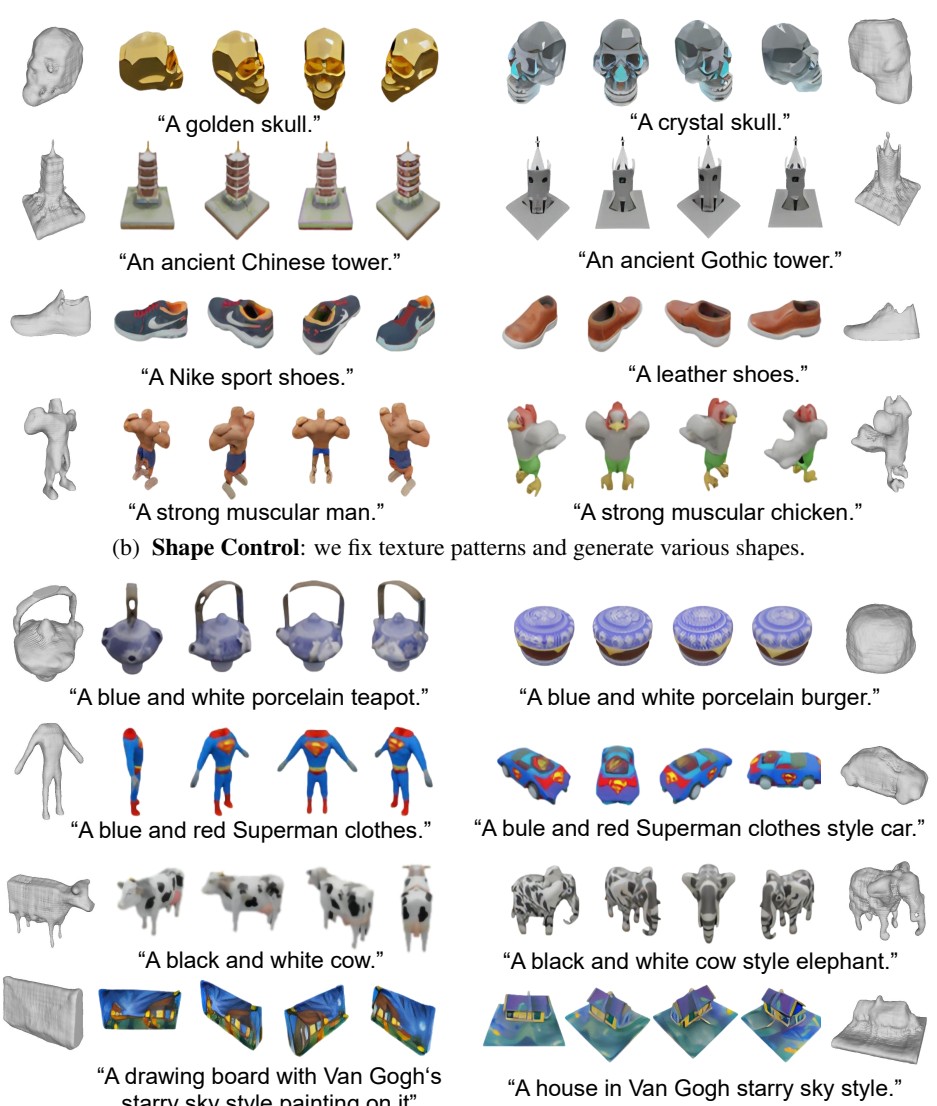

(b) **Shape Control**: we fix texture patterns and generate various shapes.

## 1 INTRODUCTION

Recent advancements in text-to-image generation (Metzer et al., 2022) have galvanized efforts to lift 2D foundational models into 3D object generation. While these methods have made strides, they predominantly focus on enriching texture quality using 2D priors, often overlooking the understanding of 3D geometry. One line of works (Poole et al., 2022; Lin et al., 2022) seeks to optimize a randomly initialized neural radiance field (NeRF) (Mildenhall et al., 2021) for 3D generation. These efforts supervise NeRF renderings using an SDS loss derived from a pre-trained 2D diffusion model while leaving behind the geometry solely restricted by the continuous density field. Without 3D constraints, these methods often lead to geometric anomalies, such as the multi-face Janus problem, and necessitate prolonged optimization periods for every single object. Liu et al. (2023a) tries to alleviate this problem by fine-tuning the 2D diffusion models on multi-view datasets, but a simple image-space multi-view correspondence constraint still falls short of ensuring genuine 3D consistency.

To ensure better 3D consistency, concurrently, some researchers proposed to directly learn 3D structures from 3D datasets (Nichol et al., 2022; Jun & Nichol, 2023). However, the majority of

objects within contemporary 3D datasets (Chang et al., 2015; Deitke et al., 2022) are synthetically generated by human designers, and their textures are typically less authentic than those of real scanned objects. Moreover, the size of 3D dataset is still an order magnitude smaller than the 2D image dataset. As a result, the trained 3D diffusion models can generate accurate 3D geometries, but they frequently produce inauthentic textures due to the limitations in 2D texture comprehension.

This motivation drives our search for an innovative method that seamlessly integrates both 3D and 2D priors within a unified framework. Still, crafting the architecture that combines 3D and 2D priors poses non-trivial challenges: i) The inherent disparity in representations between the 3D and 2D domains makes the learning of their joint distribution a non-trivial task; ii) 3D and 2D generative models are pretrained differently, potentially resulting in opposite generative directions when combined.

To conquer these challenges, we propose *Bidirectional Diffusion* (*BiDiff*), a framework that utilizes a bidirectional guidance mechanism to bridge the 3D and 2D diffusion processes. The proposed framework ensures a coordinated denoising direction across both domains. First, we anchor both 3D and 2D diffusion processes in pre-trained large 3D and 2D foundational models respectively, which ensures the robustness and versatility of both texture and geometry generation. Moreover, with the development of individual foundation models in 3D and 2D, *BiDiff* can be also continuously improved. Specifically, we use as a hybrid representation for 3D objects: the signed distance field (SDF Wang et al. (2021)) in 3D and multi-view images in 2D. With this representation, we can train a 3D diffusion model in the SDF space and a 2D diffusion model in the multi-view image space, and combine them together.

To further interconnect 3D and 2D diffusion models, we introduce bidirectional guidance to align the generative directions of them. During each diffusion step, the 2D diffusion model's output is initially incorporated into the 3D diffusion process as a 2D guidance signal. Subsequently, the rendered image produced from the 3D diffusion output is integrated into the 2D diffusion model as a 3D guidance signal. This design ensures that the two diffusion models mutually inform and adapt to one another, orchestrating a unified denoising direction.

The proposed bidirectional diffusion poses several advantages over the previous 3D generation models. First, since we introduce both the 3D and 2D diffusion models, we can modify each diffusion process independently during inference to separately control the shape and texture of the generated results, which is never achieved in previous 3D diffusion methods. As illustrated in Fig 2, we can either modulate the generated texture independently of the overall shape, or maintain consistent texture across a range of diverse shapes. Second, *BiDiff* can generate 3D objects Fig. 1(d) through a feed-forward 3D-2D joint diffusion process ($\approx$ 40s), with more authentic textures than those solely trained on 3D dataset (Jun & Nichol, 2023) Fig. 1(a), and offers explicit geometry (textured mesh) with genuine 3D consistency compared to methods (Liu et al., 2023a) that merely fine-tune 2D foundation models to acquire multi-view correspondence Fig. 1(b). Leveraging the robust capabilities of both 3D and 2D priors, the outputs generated by *BiDiff* effectively align text prompts with consistent geometrical structures. Upon generating these feed-forward results, our framework further offers an optional refinement phase, employing an optimization method (ProlificDreamer Wang et al. (2023)). This method is utilized as an efficient process for refinement, markedly enhancing processing speed (3.4h $\rightarrow$ 20min) while concurrently addressing issues of geometrical inaccuracies, such as the elimination of multi-face anomalies, as demonstrated in Fig. 1(e). In this way, our framework enables creators to rapidly adjust prompts to obtain a satisfactory preliminary 3D model through a lightweight feed-forward generation process, subsequently refining it into a high-fidelity results.

Through training on ShapeNet (Chang et al., 2015) and Objaverse 40K (Deitke et al., 2022), our framework is shown to generate high-quality textured 3D objects with strong generalizability. In summary, our contributions are as follows: 1) we propose Bidirectional Diffusion to jointly diffuse 3D and 2D in a unified framework; 2) we utilize both 3D and 2D priors to achieve a generalizable understanding of texture and geometry; 3) we can control the texture and geometry independently during sampling; 4) we utilize the outputs from *BiDiff* as a strong initialization of the optimization-based methods to improve the geometry and efficiency.

## 2 RELATED WORK

Early 3D generative methods adopt various 3D representations, including 3D voxels (Wu et al., 2016; Smith & Meger, 2017; Henzler et al., 2019), point clouds (Panos Achlioptas & Guibas,

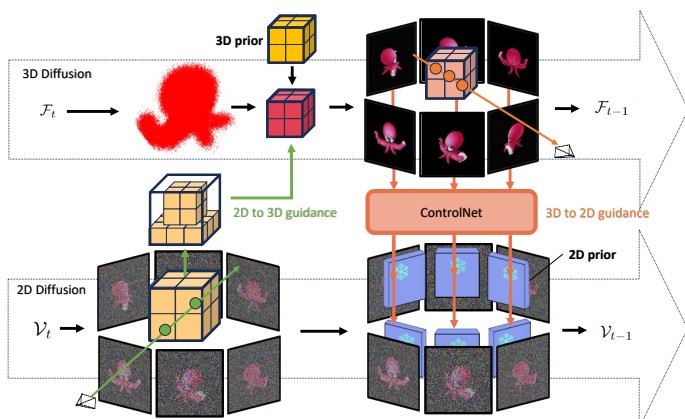

Figure 3: The framework of Bidirectional Diffusion. It jointly trains a 3D diffusion in SDF $\mathcal{F}$ space and a 2D diffusion in multi-view image $\mathcal{V}$ space, which are both enhanced by foundation models and interconnected by bidirectional guidance to achieve consistent denoising between two domains.

2018; Yang et al., 2019), meshes (Lin Gao & Zhang, 2019; Moritz Ibing & Kobbelt, 2021), and implicit functions (Chen & Zhang, 2019; Jeong Joon Park & Lovegrove, 2019) for category-level 3D generations. These methods directly train the generative model on a small-scale 3D dataset, and, as a result, the generated objects may either miss tiny geometric structures or lose diversity. Even though there are large-scale (Deitke et al., 2022) or high-quality 3D datasets (Tong Wu, 2023) in recent years, they are still much smaller than the datasets used for 2D image generation training.

With the powerful text-to-image synthesis models (Radford et al., 2021; Saharia et al., 2022; Rombach et al., 2022), a new paradigm emerges for 3D generation without large-scale 3D datasets by leveraging 2D generative model. One line of works utilizes 2D priors from pre-trained text-to-image model (known as CLIP) (Jain et al., 2022; Khalid et al., 2022) or 2D diffusion generative models (Wang et al., 2022; Lin et al., 2022; Metzer et al., 2022) to guide the optimization of underlying 3D representations. However, these models could not guarantee cross-view 3D consistency and the per-instance optimization scheme suffers both high computational cost and over-saturated problems. Later on, researchers improve these models using textual codes or depth maps (Seo et al., 2023; Deng et al., 2023; Melas-Kyriazi et al., 2023), and Wang et al. (2023) directly model 3D distribution to improve diversity. These methods alleviate the visual artifacts but still cannot guarantee high-quality 3D results.

Another line of works learn 3D priors directly from 3D datasets. As the diffusion model has been the de-facto network backbone for most recent generative models, it has been adapted to learn 3D priors using implicit spaces such as point cloud features (Zeng et al., 2022; Nichol et al., 2022), NeRF parameters (Jun & Nichol, 2023; Erkoç et al., 2023), or SDF spaces (Cheng et al., 2022; Liu et al., 2023b). The synthesized multi-view images rendered from 3D datasets were also utilized to provide cross-view 3D consistent knowledge (Liu et al., 2023a). These methods normally highlight fast inference and 3D consistent results. However, due to inferior 3D dataset quality and size, these methods generally yield visually lower-quality results with limited diversity. Recently a few methods (Qian et al., 2023; Shi et al., 2023) explored to combine 2D priors and 3D priors from individual pre-trained diffusion models, but they often suffer from inconsistent between two generative processes.

## 3 METHOD

As many previous studies Liu et al. (2023a); Qian et al. (2023) have illustrated, both 2D texture and 3D geometry are important for 3D object generation. However, incorporating 3D structural priors and 2D textural priors is challenging: i) combining both 3D and 2D generative models into a single cohesive framework is not trivial; ii) in both training and inference, two generative models may lead to opposite generative direction; iii) The scarcity of high-quality and diverse 3D data considerably hampers the generalizability of a unified 3D and 2D comprehension.

To tackle these problems, we propose *Bidirectional Diffusion*, a novel framework that marries a 3D diffusion model with a 2D diffusion model using bidirectional guidance, as illustrated in Fig. 3. For a

robust and generalizable understanding of texture and geometry, we incorporate 3D and 2D priors derived from pre-trained foundation models into their respective denoising processes. To further enhance the efficacy and ensure optimal utilization of the 3D and 2D priors while providing precise control over their influence, we present a prior enhancement strategy, which also helps to achieve decoupled texture and geometry control. Moreover, we utilize the results from *BiDiff* as a strong initialization of optimization-based methods to obtain more delicate post-optimized results efficiently. Below, we start with the introduction of bidirectional diffusion.

## 3.1 BIDIRECTIONAL DIFFUSION

To incorporate both 2D and 3D prior, we represent a 3D object using a hybrid combination of two formats: Signed Distance Field (SDF) $\mathcal{F}$ and multi-view image set $\mathcal{V} = \{\mathcal{I}^i\}_{i=1}^{M}$, where $\mathcal{F}$ is computed from signed distance values on $N \times N \times N$ grid points and $I^i$ is the $i$-th image from a multi-view image set of size $M$.

With this presentation, we learn a joint distribution $\{\mathcal{F}, \mathcal{V}\}$ utilizing two distinct diffusion models: a 3D diffusion model $\mathcal{D}_{3d}$ in the SDF space and a 2D multi-view diffusion model $\mathcal{D}_{2d}$ within the image domain. Specifically, given a timestep $t$, we add Gaussian noises to both SDF and multi-view images as

$$\mathcal{F}_t = \sqrt{\overline{\alpha}_t}\mathcal{F}_0 + \sqrt{1 - \overline{\alpha}_t}\epsilon_{3d} \text{ and } \mathcal{I}_t^i = \sqrt{\overline{\alpha}_t}\mathcal{I}_0^i + \sqrt{1 - \overline{\alpha}_t}\epsilon_{2d}^i \text{ for } \forall i, \tag{1}$$

where $\epsilon \sim \mathcal{N}(0, \mathbf{I})$ is random noise, and $\overline{\alpha}_t$ is noise schedule which is different for 3D and 2D. Subsequently, the straightforward way is to separately train these two diffusion models by minimizing the following two objectives separately:

$$L_{simple3d} = E_{\mathcal{F}_0 \sim q(\mathcal{F}_0), \epsilon_{3d} \sim \mathcal{N}(0,\mathbf{I}), t \sim U[1,T]} \|\epsilon_{3d} - \mathcal{D}_{3d}(\mathcal{F}_t, t)\|_2^2, \tag{2}$$

$$L_{simple2d} = \frac{1}{N} \sum_{i=1}^{N} (E_{\mathcal{I}_0^i \sim q(\mathcal{I}_0^i), \epsilon_{2d}^i \sim \mathcal{N}(0,\mathbf{I}), t \sim U[1,T]} \|\epsilon_{2d}^i - \mathcal{D}_{2d}(\mathcal{I}_t^i, t)\|_2^2). \tag{3}$$

However, such an approach overlooks the interplay between 3D and 2D. This oversight can lead to incongruent generation outcomes between 3D geometry and 2D multi-view images, hindering the network's capacity for concurrent 3D and 2D comprehension.

Therefore, we resolve this problem by a novel *Bidirectional Diffusion*. In this model, the consistency between 3D and 2D diffusion output is enforced through bidirectional guidance. First, we add a 2D guidance to the 3D generative process, as shown in Fig. 3. Specifically, during each denoising step $t$, we feed the previous denoised multi-view images $\mathcal{V}'_{t+1} = \{\mathcal{I}^i_{t+1}\}_{i=1}^{N}$ into the 3D diffusion model as $\epsilon'_{3d} = \mathcal{D}_{3d}(\mathcal{F}_t, \mathcal{V}'_{t+1}, t)$. This guidance steers the current 3D denoising direction to ensure 2D-3D consistency. It's worth mentioning that, during training, the denoised output $\mathcal{V}'_{t+1}$ from the previous step $t+1$ is inaccessible, therefore we directly substitute it with the ground truth $\mathcal{V}_t$. However, during inference, we utilize the denoised images from the preceding step. Then we could obtain the denoised radiance field $\mathcal{F}'_0$ given the 2D guided noise prediction $\epsilon'$ by $\mathcal{F}'_0 = \frac{1}{\sqrt{\overline{\alpha}_t}}(\mathcal{F}_t - \sqrt{1 - \overline{\alpha}_t}\epsilon'_{3d})$.

Secondly, we also add 3D guidance to the 2D generative process. Specifically, using the same camera poses, we render multi-view images $\mathcal{H}^i_t$ derived from the radiance field $\mathcal{F}'_0$ by the 3D diffusion model: $\mathcal{H}^i_t = \mathcal{R}(\mathcal{F}'_0, \mathcal{P}^i), i = 1, ...N$. These images are further used as a guidance to the 2D multi-view denoising process $\mathcal{D}_{2d}$, realizing the 3D-to-2D guidance: $\epsilon'_{2d} = \mathcal{D}_{2d}(\mathcal{V}_t, \{\mathcal{H}^i_t\}_{i=1}^{N}, t)$.

In this manner, we can seamlessly integrate and synchronize both the 3D and 2D diffusion processes within a unified framework. In the following sections, we will delve into each component in detail.

## 3.2 3D DIFFUSION MODEL

Our 3D diffusion model aims to generate a neural surface field (NeuS) Long et al. (2022), with novel 2D-to-3D guidance derived from the denoised 2D multi-view images. To train our 3D diffusion model, at each training timestep $t$, we add noise to a clean radiance field, yielding the noisy radiance field $\mathcal{F}_t$. This field, combined with the $t$ embeddings and the text embeddings, is then passed through 3D sparse convolutions to generate a 3D feature volume $\mathcal{M}$ as: $\mathcal{M} = \text{Sparse3DConv}(\mathcal{F}_t, t, \text{text})$. Simultaneously, using the denoised multi-view images $\mathcal{V}'_{t+1}$ from the previous step of the 2D diffusion model, we project the $N \times N \times N$ grid points from $\mathcal{M}$ onto all the $M$ views. For each grid point $p$, we aggregate the image features into 3D space by calculating the mean and variance of the $N$ interpolated features, yielding the image-conditioned feature volume $\mathcal{N}$:

$$\mathcal{N}(p) = [\text{Mean}(\mathcal{V}'_{t+1}(\pi(p))), \text{Var}(\mathcal{V}'_{t+1}(\pi(p)))], \tag{4}$$

where $\pi$ denotes the projection operation from 3D to 2D image plane. For the setting of not using 3D priors, we fuse these two feature volumes with further sparse convolutions and predict the clean $\mathcal{F}_0$ using the fused features.

One important design of our 3D diffusion model is that it incorporates geometry priors derived from the 3D foundation model, Shap-E (Jun & Nichol, 2023). Shap-E is a latent diffusion (Metzer et al., 2022) model trained on several millions 3D objects, and thus ensures the genuineness of generated 3D objects. Still, we do not want to limit the creativity of our 3D generative model by Shap-E model, and still maintain the capability of generating novel objects that Shap-E cannot.

To achieve this target, We design a feature volume $\mathcal{G}$ to represent a radiance field converted from the latent code $\mathcal{C}$. It is implemented using NeRF MLPs by setting their parameters to the latent code $\mathcal{C}$: $\mathcal{G}(p) = \mathrm{MLP}(\lambda(p); \theta = \mathcal{C})$, where $\lambda$ denotes the positional encoding operation.

Still, one limitation of the usage of Shap-E latent code is that the network is inclined to shortcut the training process, effectively memorizing the radiance field derived from Shap-E. To generate 3D objects beyond Shap-E model, we add Gaussian noise at level $t_0$ to the clean latent code, resulting in the noisy latent representation $\mathcal{C}_{t_0}$, where $t_0$ represents a predefined constant timestep. Subsequently, the noisy radiance field $\mathcal{G}_{t_0}$ is decoded by substituting $\mathcal{C}$ with $\mathcal{C}_{t_0}$. This design establishes a coarse-to-fine relationship between the 3D prior and the ground truth, prompting the 3D diffusion process to leverage the 3D prior without becoming excessively dependent on it.

In this way, we can finally get the fused feature volumes by:

$$\mathcal{S} = \mathcal{U}([\mathcal{M}, \mathrm{Sparse3DConv}(\mathcal{N}), \mathrm{Sparse3DConv}(\mathcal{G}_{t_0})]), \qquad (5)$$

where $\mathcal{U}$ denotes 3D sparse U-Net. Then we can query features from $\mathcal{S}$ for each grid point $p$ and decode it to SDF values through several MLPs: $\mathcal{F}'_0(p) = \mathrm{MLP}(\mathcal{S}(p), \lambda(p))$, where $\mathcal{S}(p)$ represents the interpolated features from $\mathcal{S}$ at position $p$. Our experiments also demonstrate that our model can generate 3D objects beyond Shap-E model.

### 3.3 2D DIFFUSION MODEL

Our 2D diffusion model simultaneously generates multi-view images, by jointly denoise multi-view noisy images $\mathcal{V}_t = \left\{ \mathcal{I}_t^i \right\}_{i=1}^{M}$. To encourage 2D-3D consistency, the 2D diffusion model is also guided by the 3D radiance field output from 3D diffusion process mentioned above. Specifically, for better image quality, we build our 2D multi-view diffusion model on the basis of several independently frozen foundation models (e.g., DeepFloyd) to harness the potent 2D priors. Each of these frozen 2D foundation models is modulated by view-specific 3D-consistent residual features and responsible for the denoising of a specific view, as described below.

First, to achieve 3D-to-2D guidance, we render multi-view images from the 3D denoised radiance field $\mathcal{F}'_0$. Note that the radiance field consists of a density field and a color field. The density field is constructed from the signed distance field (SDF) generated by our 3D diffusion model using S-density introduced in NeuS (Wang et al., 2021). To obtain the color field, we apply another color MLP to the feature volume in the 3D diffusion process.

Upon obtaining the color field $c$ and density field $\sigma$, we conduct volumetric rendering on each ray $\boldsymbol{r}(m) = \boldsymbol{o} + m\boldsymbol{d}$ which extends from the camera origin $\boldsymbol{o}$ along a direction $\boldsymbol{d}$ to produce multi-view consistent images $\left\{ \mathcal{H}^i \right\}_{i=1}^{M}$:

$$\hat{C}(\boldsymbol{r}) = \int_0^{\infty} T(m)\sigma(\boldsymbol{r}(m)))c(\boldsymbol{r}(m)), \boldsymbol{d})dm, \qquad (6)$$

where $T(m) = \exp(-\int_0^m \sigma(\mathrm{r(s)})\mathrm{ds})$ handles occlusion.

Secondly, we use these rendered multi-view images as guidance for the 2D foundation model. We first use a shared feature extractor $\mathcal{E}$ to extract hierarchical multi-view consistent features from these images. Then each extracted features are added as residuals to the decoder of its corresponding frozen 2D foundation denoising U-Net, achieving multi-view modulation and joint denoising following ControlNet (Zhang & Agrawala, 2023): $\hat{\boldsymbol{f}}_k^i = \boldsymbol{f}_k^i + \mathrm{ZeroConv}(\mathcal{E}(\mathcal{H}^i)[k])$, where $\boldsymbol{f}_k^i$ denotes the original feature maps of the $k$-th decoder layer in 2D foundation model, $\mathcal{E}(\mathcal{H}^i)[k]$ denotes the $k$-th residual features of the $i$-th view, and ZeroConv (Zhang & Agrawala, 2023) is $1 \times 1$ convolution which is initialized by zeros and gradually updated during training. Experimental results show that this 3D-to-2D guidance helps to ensure multi-view consistency and facilitate geometry understanding.

### 3.4 Prior Enhancement Strategy

In addition to bidirectional guidance, we also propose a prior enhancement strategy to empower a manual control of the strength of 3D and 2D priors independently, which offers a natural mechanism to encourage decoupled texture and geometry control. Inspired by the classifier-free guidance (Ho & Salimans, 2022), during training, we randomly drop the information from 3D priors by setting condition feature volume from $\mathcal{G}$ to zero and weaken the 2D priors by using empty text prompts. Consequently, upon completing the training, we can employ two guidance scales, $\gamma_{3d}$ and $\gamma_{2d}$, to independently modulate the influence of these two priors.

Specifically, to adjust the strength of 3D prior, we calculate the difference between 3D diffusion outputs with and without conditional 3D feature volumes, and add them back to 3D diffusion output:

$$\hat{\epsilon}_{3d} = \mathcal{D}_{3d}(\mathcal{F}_t, \mathcal{V}'_{t+1}, t) + \gamma_{3d} \cdot ((\mathcal{D}_{3d}(\mathcal{F}_t, \mathcal{V}'_{t+1}, t | \mathcal{G}) - \mathcal{D}_{3d}(\mathcal{F}_t, \mathcal{V}'_{t+1}, t)). \tag{7}$$

Then we can control the strength of 3D prior by adjusting the weight $\gamma_{3d}$ of this difference term. When $\gamma_{3d} = 0$, it will completely ignore 3D prior. When $\gamma_{3d} = 1$, this is just the previous model that uses both 3D prior and 2D prior. When $\gamma_{3d} > 1$, the model will produce geometries close to the conditional radiance field but with less diversity.

Also, we can similarly adjust the strength of 2D priors by adding differences between 2D diffusion outputs with and without conditional 2D text input:

$$\hat{\epsilon}_{2d} = \mathcal{D}_{2d}(\mathcal{V}_t, \{\mathcal{H}_t^i\}_{i=1}^M, t) + \gamma_{2d} \cdot ((\mathcal{D}_{2d}(\mathcal{V}_t, \{\mathcal{H}_t^i\}_{i=1}^M, t | text)) - \mathcal{D}_{2d}(\mathcal{V}_t, \{\mathcal{H}_t^i\}_{i=1}^M, t)). \tag{8}$$

Increasing $\gamma_{2d}$ results in more coherent textures with text, albeit at the expense of diversity. It is worth noting that while we adjust the 3D and 2D priors independently via Eq. (7) and Eq. (8), the influence inherently propagates to the other domain due to the intertwined nature of our bidirectional diffusion process.

To achieve separate texture and geometry control, on the one hand, we first fix the initial 3D noisy SDF grids and fix the conditional radiance field $\mathcal{C}_{t_0}$ while enlarging its influence by Eq. (7). In this way, we can modify the 2D diffusion process by adjusting the text prompts to change texture while maintaining overall shapes; on the other hand, we can keep the texture styles by maintaining keywords in text prompts and enlarge its influence by Eq. (8), then the shape can be adjusted by modifying the 3D diffusion process like varying conditional radiance field.

### 3.5 Optimization with BiDiff Initialization

Once obtain the denoised radiance field $\mathcal{F}_0$, we can further use it as a strong initialization of the optimization-based methods for further refinement. Importantly, since our generated $\mathcal{F}_0$ has text-aligned textures derived from the powerful 2D prior as well as accurate geometry guided by the 3D prior, the optimization started from this strong initialization can be rather efficient ($\approx$ 20min) and avoid incorrect geometries like multi-face and floaters.

Specifically, we first convert our radiance field $\mathcal{F}_0$ into a higher resolution radiance field $\overline{\mathcal{F}}_0$ that supports $512 \times 512$ resolution image rendering. This process is achieved by a fast NeRF distillation operation ($\approx$ 2min), which first bounds the occupancy grids of $\overline{\mathcal{F}}_0$ with the estimated binary grids (transmittance > 0.01) from $\mathcal{F}_0$, then overfit $\overline{\mathcal{F}}_0$ to $\mathcal{F}_0$ by simultaneously optimizing its downsampled density field and interpolated random view renderings with $L_1$ loss between the corresponding results in $\mathcal{F}_0$. Thanks to this flexible and fast distillation operation, we can initialize our generated radiance field into any optimization-based method efficiently without the need to match its 3D representation. In our experiments, we use the InstantNGP Müller et al. (2022) as the high-resolution radiance field.

After initialization, we optimize $\overline{\mathcal{F}}_0$ by SDS loss following the previous methods Poole et al. (2022); Wang et al. (2023). It is noteworthy that since we already have a good initialized radiance field, we do not need to apply a large noise level SDS loss as in previous methods. In contrast, we set the ratio range of denoise timestep to [0.02, 0.5] during the entire optimization process.

## 4 Experiment

In this section, we described our experimental results. We train our framework on the ShapeNet-Chair (Chang et al., 2015) and Objaverse LVIS 40k datasets (Deitke et al., 2022). We use the pre-trained DeepFloyd-IF-XL as our 2D foundation model and use Shap-E (Jun & Nichol, 2023) as our 3D priors. We adopt the SparseNeuS (Long et al., 2022) as the Neural Surface Field with $N = 128$. We follow ControlNet (Zhang & Agrawala, 2023) and render $M = 8$ multi-view images

with $64 \times 64$ resolution from SparseNeuS to implement the 3D-to-2D guidance. We train our framework on 4 NVIDIA A100 GPUs for both ShapeNet and Objaverse 40k experiments with batch size of 4. During sampling, we set the 3D and 2D prior guidance scale to 3.0 and 7.5 respectively. More details including the data processing and model architecture can be found in the appendix. We discuss the evaluation and ablation study results below.

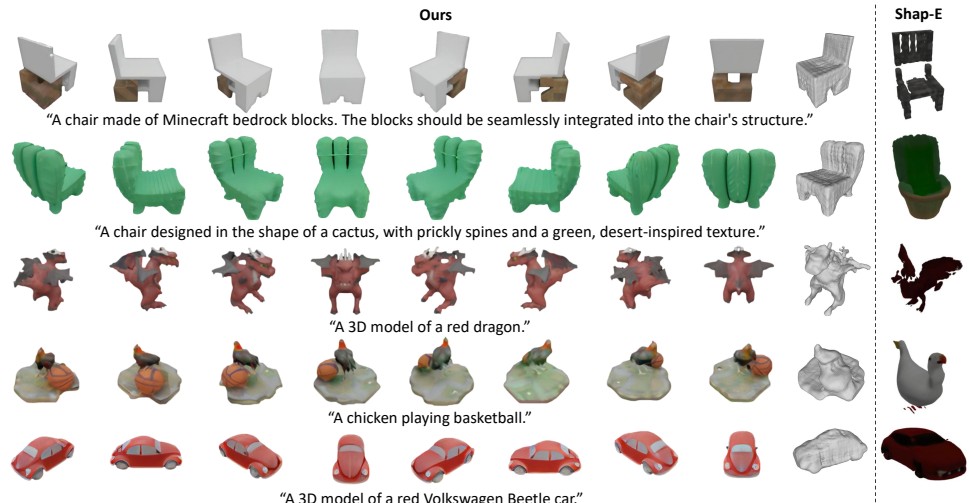

Figure 4: Qualitative sampling results of Bidirectional Diffusion model, including multi-view images and 3D mesh from diffusion sampling. The top two lines are the results on the Shapenet-Chair, and the bottom three lines are the results on the Objaverse. We compared the results of Shap-E in the last column.

## 4.1 TEXT-TO-3D RESULTS

**ShapeNet-Chair and Objaverse results.** The first and second rows of Fig. 4 present our results trained on the ShapeNet-Chair dataset. Even the chair category frequently exhibits intricate geometric details, our framework demonstrates adeptness in capturing fine geometries. The bottom three rows of Fig. 4 show a capacity to craft a plethora of 3D objects that closely adhere to the given textual prompts. This reaffirms the hypothesis that our method learns a generalizable comprehension of both texture and geometry.

**Decouple geometry and texture control.** At last, we illustrate that our Bidirectional Diffusion separately control geometry generation through 3D diffusion model and texture generation through the 2D diffusion model. This is the first work that enables separate controls in diffusion process. First, as illustrated in Fig. 2(a), when the 3D prior is fixed, we have the flexibility to manipulate the 2D diffusion model using varying textual prompts to guide the texture generation process. This capability enables the generation of a diverse range of textured objects while maintaining a consistent overall shape. Second, when we fix the textual prompt for the 2D priors (e.g., "a xxx with Van Gogh starry sky style"), adjustments to the 3D diffusion model can be made by varying the conditional radiance field derived from the 3D priors. This procedure results in the generation of a variety of shapes, while maintaining a similar texture, as shown in Fig. 2(b).

## 4.2 COMPARED WITH OTHER METHODS

Table 1: CLIP R-precision.

| Method | R-P | time |
|---|---|---|
| DreamFusion | 0.67 | 1.1h |
| ProlificDreamer | 0.83 | 3.4h |
| Ours-sampling | 0.79 | **40s** |
| Ours-post | **0.85** | 20min |

**Compared with DreamFusion Series.** Our framework is capable of simultaneously generating multi-view consistent images alongside a 3D mesh in a scalable manner, contrasting with the Dreamfusion (Poole et al., 2022) series which relies on a one-by-one optimization approach. Table 1 reports the CLIP R-Precision (Jun & Nichol, 2023) and inference time on 50 test prompts (manually derived from the captioned objaverse test) to quantitively evaluate these methods. Dreamfusion requires 1 hour to generate a single object. ProlificDreamer (Wang et al., 2023) improves the texture quality, but at the expense of extended optimization time, taking approximately 3.4 hours and leading to more severe multi-face problems. In contrast, our method can produce realistic textured objects with reasonable geometry in 40 seconds. Furthermore, *BiDiff* can serve as a strong prior for optimization-based methods and significantly boost their performance. Initializing the radiance field in ProlificDreamer

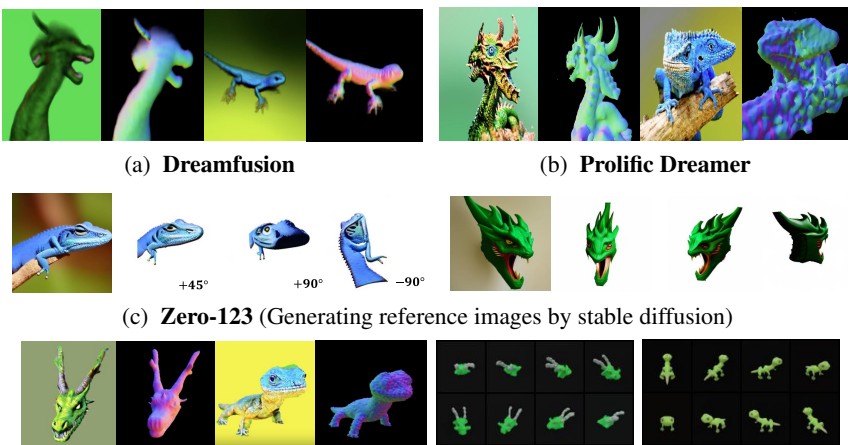

(a) **Dreamfusion**  (b) **Prolific Dreamer**

(c) **Zero-123** (Generating reference images by stable diffusion)

(d) **BiDiff (Ours)**

Figure 5: Comparison with other optimization or multi-view diffusion based works. The text prompts are "a green dragon head" and "a cute lizard". We show both multi-view images (right of $(d)$) and refined results (left of $(d)$).

with our outputs, shows remarkable improvements in both quality and computational efficiency, as shown in Fig. 5.

**Compared with Zero-123 Series** Given one reference image, Zero-123 (Liu et al., 2023a) produces images from novel viewpoints by fine-tuning a pre-trained 2D diffusion model on multi-view datasets. However, this method employs cross-view attention to establish multi-view correspondence without an inherent un-

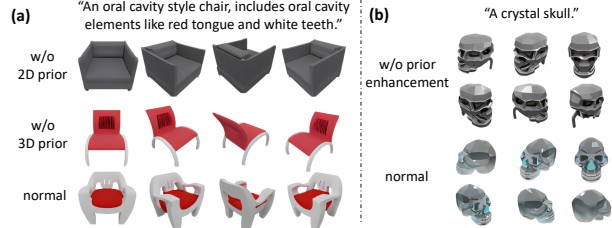

Figure 6: Results for ablation experiments.

derstanding of 3D structures, inevitably leading to inconsistent multi-view images as shown in Fig. 5. Moreover, the Zero-123 series can not directly generate the 3D mesh, requiring substantial post-processing (SDS loss) to acquire the geometry. In contrast, our framework ingeniously incorporates 3D priors and achieves 3D geometry understanding within a 3D diffusion process. This design enables the simultaneous generation of multi-view consistent images and a 3D mesh, as illustrated in Fig. 2.

### 4.3 ABALATION STUDIES

We perform comprehensive ablation studies to evaluate the importance of each component. More ablation results can be found in the appendix.

**3D priors.**  To assess the impact of 3D priors, we eliminate the conditional radiance field from Shap-E and solely train the 3D geometry from scratch. The experimental results in Fig. 6 (a) demonstrate that in the absence of the 3D priors, our framework can only generate the common objects in the training set.

**2D priors**  To delve into the impact of 2D priors, we randomly initiate the parameters of 2D diffusion model, instead finetuning on a pretrained model. The results in Fig. 6 (a) shows that in the absence of 2D priors, the textures generated tend to fit the stylistic attributes of the synthetic training data. Conversely, with 2D priors, we can produce more realistic textures.

**Prior enhancement strategy**  As discussed in Section 3.4, we can adjust the influence of both 3D and 2D priors by the prior enhancement strategy. Fig. 6 (b) shows the results of different enhancement extent under different scale factors. It shows that the prior enhance meant strategy plays a vital role in achieving decoupled texture and geometry control.

### 5 CONCLUSION

In this paper, we propose Bidirectional Diffusion, which incorporates both 3D and 2D diffusion processes into a unified framework. Furthermore, Bidirectional Diffusion leverages the robust priors from 3D and 2D foundation models, achieving generalizable geometry and texture understanding.

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
