# A  APPENDIX

In the supplementary material, we first introduce the data processing pipeline in (§ A.1), then provide more implementation details of the model architecture (§ A.2), more training details in (§ A.3), and give more ablation results in (§ A.4).

## A.1  DATA PROCESSING

As mentioned in the main paper, we use 6k ShapNet-Chair Chang et al. (2015) and LVIS Objaverse 40k Deitke et al. (2022) as our training datasets. We obtain the Objaverse 40k dataset by filtering objects with LVIS category labels in the 800k Objaverse data. To process data for the 2D diffusion process, we render each 3D object into 8 images with a fixed elevation of $30°$ and evenly distributed azimuth from $-180°$ to $180°$ by blender. These fixed view images serve as the ground truth multi-view image set $\mathcal{V}$. In addition, we also randomly render 16 views to supervise the novel view rendering of the denoised radiance field $\mathcal{F}'_0$. All the images are rendered at a resolution of $256 \times 256$. Since we adopt the DeepFloyd as our 2D foundation model which runs at a resolution of $64 \times 64$, the rendered images are downsampled to $64 \times 64$ during training. To process data for the 3D diffusion, we compute the signed distance of each 3D object at each $N \times N \times N$ grid point within a $[-1, 1]$ cube, where $N$ is set to 128 in our experiments. To obtain the latent code $\mathcal{C}$ for each object, we use the encoder in Shap-E Jun & Nichol (2023) to encode each object and apply $t_0 = 0.4$ level Gaussian noise to $\mathcal{C}$ to get noisy $\mathcal{C}_{t_0}$, and then decode the condition radiance field during training.

Furthermore, both the ShapNet-Chair and Objaverse dataset contains no text prompts, so we use Blip-2 Li et al. (2023) to generate labels for the objaverse object by rendering the image from a positive view. For evaluation, we manually choose 50 text prompts from the Objaverse dataset without LVIS label, ensuring the text prompts have not been trained during training.

## A.2  MODEL ARCHITEXTURE DETAILS

Our framework contains a 3D denoising network built upon 3D SparseConv U-Net and a 2D denoising network built upon 2D U-Net. Below we provide more details for each of them.

### A.2.1  3D DENOISING NETWORK

Given the input feature volume $\mathcal{S}_{\text{in}} = \text{Concat}(\mathcal{M}, \text{Sparse3DConv}(\mathcal{N}), \text{Sparse3DConv}(\mathcal{G}_{t_0}))$ as discussed in Section 3.2 of the main paper, we use a 3D sparse U-Net $\mathcal{U}$ to denoise the signed distance field. Specifically, we first use a $1 \times 1 \times 1$ convolution to adjust the input channels to 128. Then we stack four $3 \times 3 \times 3$ sparse 3D convolution blocks to extract hierarchical features while obtaining downsampled $8 \times 8 \times 8$ feature grids. It is noteworthy that we inject the timestep and text embeddings into each sparse convolution block to make the network aware of the current noise level and text information. In practice, we first use an MLP to project the scalar timestep $t$ to high-dimensional features and fuse it with the text embeddings with another MLP to get the fused embeddings as follows:

$$\text{emb} = \text{MLP}_2(\text{Concat}(\text{emb}_{\text{text}}, \text{MLP}_1(t))), \tag{1}$$

where $\text{emb}_{\text{text}}$ denotes the text embeddings. Then in each sparse convolution block, we project the fused embeddings to scale $\beta$ and shift $\gamma$:

$$\beta, \gamma = \text{Chunk}(\text{MLP}_{\text{proj}}(\text{GeLU}(\text{emb}))), \tag{2}$$

where GeLU is activated function, Chunk operation splits the projected features into two equal parts along the channel dimension. After that, we introduce modulation to the sparse convolution by:

$$\mathcal{S}_{k+1} = (1 + \beta)(\text{SparseConv}(\text{GroupNorm}(\mathcal{S}_k))) + \gamma, \tag{3}$$

where k denotes the feature level, $\mathcal{S}_k$ and $\mathcal{S}_{k+1}$ are the input and output of the $k$-th level sparse convolution block. Subsequently, we use 4 sparse deconvolution blocks to upsample the bottleneck feature grids with residuals linked from the extracted hierarchical features:

$$\mathcal{S}'_k = \text{SparseDeConv}(\mathcal{S}'_{k+1}) + \mathcal{S}_k, \tag{4}$$

where $\mathcal{S}'_{k+1}$ and $\mathcal{S}'_k$ are the input and output of the $k$-th level sparse de-convolution block, and obtain the output features $\mathcal{S}$ of the 3D U-Net.

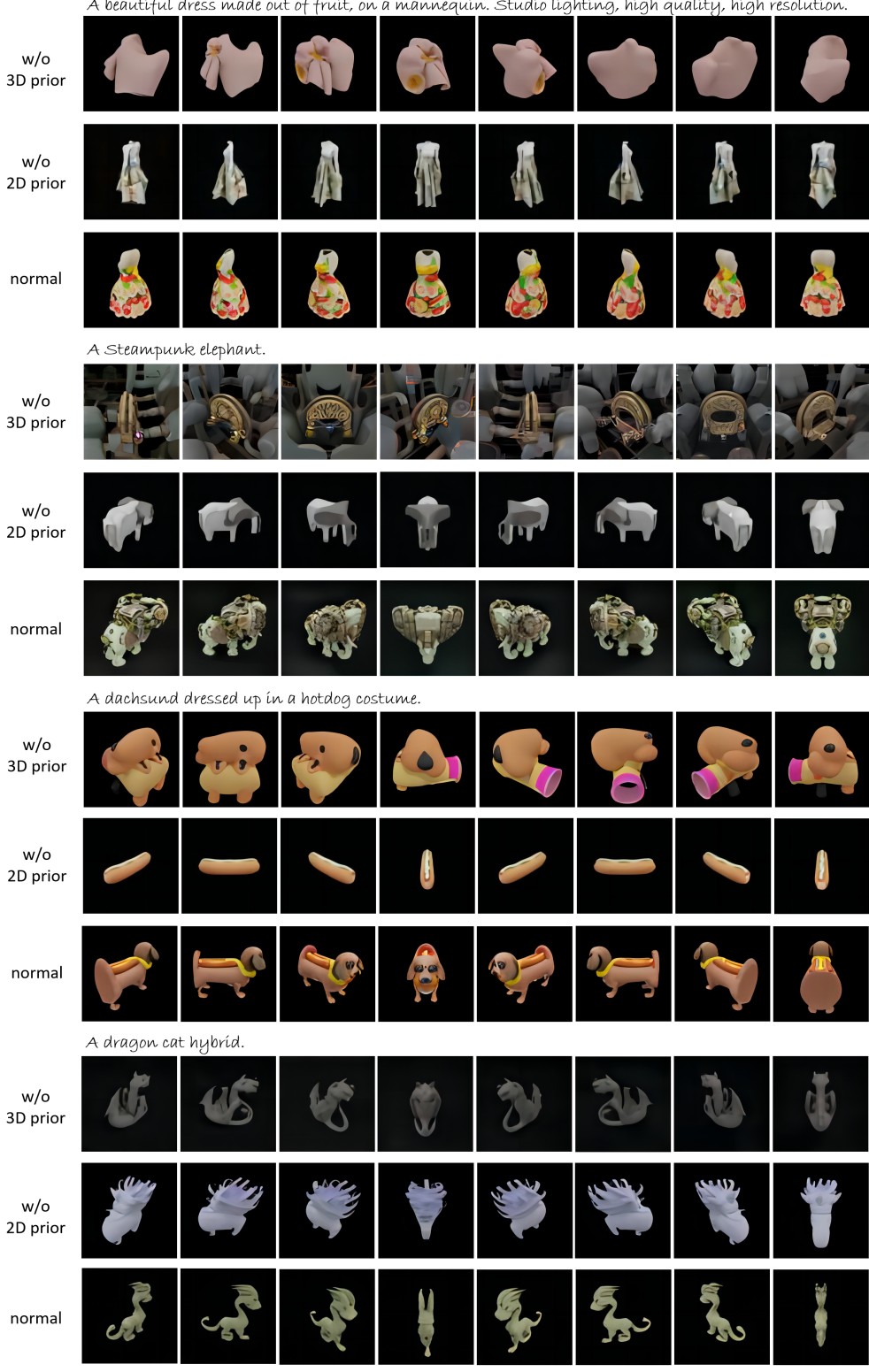

Figure 1: More ablation results of 2D, 3D prior.

To obtain the denoised signed distance field, we first query each 3D position $p$ in the fused feature grid $\mathcal{S}$ to fetch its feature $\mathcal{S}(p)$ by Trilinear Interpolation. Then we apply several MLPs (we adopt the ResNetFC blocks in Yu et al. (2021) to predict the signed distance at position $p$:

$$\mathcal{F}_0' = \text{MLP}(\mathcal{S}(p), \lambda(p)), \tag{5}$$

where $\lambda(p)$ is the positional encoding:

$$\lambda(p) = (\sin(2^0 \omega p), \cos(2^0 \omega p), \sin(2^1 \omega p), \cos(2^1 \omega p), ..., \sin(2^{L-1} \omega p), \cos(2^{L-1} \omega p)). \tag{6}$$

$L$ is set to 6 in all experiments.

### A.2.2  2D DENOISING NETWORK

Our 2D denoising network contains a U-Net of the 2D foundation model (DeepFloyd) and a Control-Net Zhang & Agrawala (2023) modulation module to jointly denoise the multi-view image set. In practice, given the $M$ noisy images $\mathcal{V}_t = \left\{ \mathcal{I}_t^i \right\}_{i=1}^M$ from the 2D diffusion process and $M$ rendered images $\left\{ \mathcal{H}^i \right\}_{i=1}^M$ from the 3D diffusion process as mentioned in Section 3.3 of the main paper, we first reshape both of them from $[B, M, C, H, W]$ to $[B \times M, C, H, W]$, where $B, C, H, W$ denotes batch size, channel, height, width, respectively. Then we feed the noisy images to the frozen encoder $\mathcal{E}^*$ of DeepFloyd to get encoded features:

$$P = \mathcal{E}^*(\text{Reshape}(\left\{ \mathcal{I}_t^i \right\}_{i=1}^M), t, \text{emb}_{\text{text}}). \tag{7}$$

$P = \left\{ p^k \right\}_{k=1}^K$ where $p^k$ denotes the $k$-th features of the total $K$ feature levels. Simultaneously, we feed the rendered images to the trainable copy encoder $\mathcal{E}$ of ControlNet to obtain the hierarchical 3D consistent condition features:

$$Q = \mathcal{E}(\text{Reshape}(\left\{ \mathcal{H}^i \right\}_{i=1}^M), t, \text{emb}_{\text{text}}), \tag{8}$$

where $Q = \left\{ q^k \right\}_{i=1}^K$. Subsequently, we decode $P$ with the frozen decoder $\mathcal{D}^*$ of DeepFloyd and the condition residual features $Q$. Specifically, in the $k$-th decoding stage, we first apply zero-convolutions to the condition feature $q^k$ and then add it to the original decoded features as residuals:

$$\hat{f}^k = p^k + \mathcal{D}_{k-1}^*(p^{k-1}) + \text{ZeroConv}(q^k), \tag{9}$$

where $\mathcal{D}_{k-1}^*$ denotes the $k-1$-th frozen decoding layer of DeepFloyd. In this way, we can denoise the multi-view noisy images in a unified manner by introducing the 3D consistent condition signal as guidance. In practice, we set $M = 8$ in our experiments.

### A.3  MORE TRAINING DETAILS

We train our framework on 4 NVIDIA A100 GPUs with a batch size of 4. For ShapeNet-Chair, the training takes about 8 hours to converge. For Objaverse 40k, the training takes 5 days. We use the AdamW optimizer with $\beta = (0.9, 0.999)$ and weight decay $= 0.01$. Notably, we set the learning rate of the 2D diffusion model to $2 \times 10^{-6}$ while using a much larger learning rate of $5 \times 10^{-5}$ for the 3D diffusion model.

### A.4  MORE ABLATION RESULTS

In this section, we provide additional results and analysis for the ablation of 3D and 2D priors in Fig. 1.