# OpenReview forum: "Text-to-3D Generation with Bidirectional Diffusion using both 3D and 2D priors"
_ICLR.cc/2024/Conference — ICLR 2024 Conference Withdrawn Submission_

### Official Review · Reviewer_Ymwx · 2023-10-28

**Soundness:** 2 fair
**Presentation:** 2 fair
**Contribution:** 2 fair
**Rating:** 3
**Confidence:** 4

**Summary:**

This paper proposes a method for a 2D diffusion and a 3D diffusion model to exchange information during inference so that they can complement each other. It adds additional conditioning branches to two pretrained 2D and 3D diffusion models, and allow them to condition on each other at every step of the diffusion inference process. The model can generate 3D results more faithful to text than simply using a standalone 3D diffusion model, and enables disentangled control of geometry and texture.

**Strengths:**

1. The idea of allowing 2D and 3D diffusion models to condition on each other's outputs at every inference step is novel. This method can be used with strong independently pre-trained 3D and 2D diffusion backbones, and so has the potential to incorporate the rich prior in both models into one.

2. The intrinsic capability of independently controlling texture and geometry is new compared to previous text-to-3D works as far as I know.

**Weaknesses:**

The main weaknesses of this paper are the lack of enough qualitative results and the ambiguity of explanation.

1. In the ablation study of 4.3, only one particular qualitative example is shown to demonstrate the effectiveness of different components. This is far from being convincing. The authors should have included more than 10 results of different prompts in the appendix for that.

2. In the "bidirectional guidance" part of section 4.3 Ablation Studies, the results shown at the top row of figure 6 seem to be totally different shapes. I understand this can happen for the 2D diffusion model. However the text also says "... and the 3D diffusion model manifests anomalies in both texture and geometric constructs.". But where are the 3D diffusion results? From my understanding the results from the 3D diffusion model should always look like the same shape and yield consistent multi-view renderings. I did not find these results in figure 6.

3. Figure 4 shows the main qualitative results of the proposed feed-forward method. However there is no comparison to previous methods. I think at least the comparison to Shap-E should be included.

4. The results of Zero-1-to-3 shown in figure 5 are weird. Why all the others methods shown are the final 3D results with mesh visualization, but the Zero-1-to-3 only has multi-view generation results? My understanding is to generation the 3D results at the left lower corner of Figure 5, we still need to use the SDS loss. If this is true, then a directly competitor should be using Zero-1-to-3 with SDS loss.

5. More details about the decoupled geometry and texture control in page 8 are needed. What does it mean to fix the 3D prior? Do you mean fixing the initial noise of the 3D diffusion? When fixing the textual prompt if the 2D diffusion, do you also fix the initial noise?

**Questions:**

1. In the "Results after optimization" section of the website the authors show results of combining the proposed method and SDS optimization. Why the multi view images generated on the left are low quality and lack details, while the 3D results on the right suddenly have much richer texture and geometry?

2. During training, since we do not have the results of the intermediate denoised results for any particular timestep t, the original GT 3D shape and multi-view images are used as the conditioning signals. I feel this domain gap seems to be a serious problem. Do the authors have any thoughts on eliminating this gap?

---

> ### Author Response · Authors · 2023-11-16
> **Reply to Reviewer Ymwx**
>
> We sincerely thank the reviewer for providing thoughtful review and positive feedback, and will update the paper accordingly. Below are our responses to the main questions by the reviewer.
>
> **1. Problem of separation of shape and texture control.**
>
> Please refer to Question 1 in interpretation of common questions.
>
> **2. Problem of results in websit.**
>
> Sorry for any confusion caused by our video. The results on the left are from the bidirectional diffusion feed-forward sampling, while those on the right are after applying SDS optimization to these results. Please refer to Question 1 in common questions about the gap between the optimized results and the results before optimization.
>
> **3. The domain gap of conditioning signals.**
>
> During training, the conditioning signals for the 3D network are noisy multi-view images, not the clean ground-truth ones. At inference time, these conditioning signals are also noisy images, resulting from the previous 2D diffusion step. Therefore, there is no training and inference gap. The conditioning signals for the 2D network are the multi-view images rendered by the 3D network, which are also consistent with inference.
>
> Moreover, using noisy multi-view images does not lead to significant problems. In the early stages of sampling, when noise is relatively high, the 3D network's denoising relies more on the 3D prior, and at this time, the noise-induced uncertainty in 3D structure is acceptable. In the later stages of sampling, the smaller noise hardly affects the generation of 3D structure. Therefore, throughout the entire sampling process, the 2D and 3D representations are denoised synchronously, with bidirectional adjustment at each step to ensure consistency between 2D and 3D.
>
> **4. More results and comparisons.**
>
> We supplemented more results in revision main paper and supplementary material.

---

### Official Review · Reviewer_ZH4m · 2023-10-29

**Soundness:** 3 good
**Presentation:** 2 fair
**Contribution:** 3 good
**Rating:** 6
**Confidence:** 4

**Summary:**

This paper presents a new 3D generation method by combining a 3D diffusion model and a 2D diffusion model via bidirectional guidance. The 3D diffusion model is based on the NeuS representation and incorperates geometric priors from the large scale Shap-E model. The 2D diffusion model is finetuned from DeepFloyd-IF for multi-view image generation. During each diffusion step, the 3D diffusion model takes image features from the prediction of 2D diffusion model, and the 2D diffusion model is further conditioned on the multi-view renderings from the prediction of 3D diffusion model. In this manner, the two models are "synchonized" to achieve corherent 3D generation. Compared to previous optimization-based methods, the proposed approach has a lower generation time and the outputs suffer from less multi-face Janus problem.

**Strengths:**

Bridging 3D and 2D diffusion models seems novel. There are mainly two types of 3D generation methods in recent years, one that learns a 3D generator directly from 3D datasets (e.g., Shape-E) and the other that distill a 3D object from 2D generative models (e.g., DreamFusion). The former usually generates 3D shapes with good geometry but often doesn't generalize well due to the limited 3D dataset. The later leverages the genealization ability of large scale 2D image model but often suffer from inaccurate geometry (e.g., Janus problem).

This work takes a step forward by combining the advantage of both --- prior from 3D dataset helps with the geometry and prior from 2D images improves the texture quality.

The results on the webpage look quite decent.

**Weaknesses:**

- The paper claims to be "the first 3D generative model that can separately control texture and geometry generation". I think this is a bit over-claiming. There are many prior works that can separately control the generation of texture and geometry. For example, TM-Net[1] and GET3D[2]. I would suggest removing such a claim.
- Results with or without post optimization should be labeled more clearly. The video results on the webpage look quite good, but figure 2 results seem to be of lower quality and resolution. I'm a bit confused by the results. I suppose the difference is whether to use post optimization. If so, such difference should be illustrated more clearly, so that we know how much the proposed diffusion model really contribute.
- The paper seems to be rushed, and there are many typos, including
  - page 3, "As result, ..." -> "As a result, ...", and on the same line, "..., they frequently produce ..." -> "..., but they frequently produce ..."
  - page 3, "... which ensures ensures ..." -> "... which ensures ..."
  - page 5, "Fig.[]." not linked
  - page 5, "the straightforward way to separately ..." -> "the straightforward way is to separately..."

[1] TM-NET: Deep Generative Networks for Textured Meshes, SIGGRAPH 2021
[2] GET3D: A Generative Model of High Quality 3D Textured Shapes Learned from Images, NeurIPS 2022

**Questions:**

Some necessary details are missing.

- As described in Sec. 3.3., the 2D diffusion model jointly denoise multi-view images. How does it work exactly? Are these multi-view images just stacked along the channel dimension? If so, what's the ordering?
- How does post-optimization work? For example, what is the optimization objactive and guidance weight? It looks like the post-optimization improves the final quality a lot, so I think it's necessary to include all the details.

---

> ### Author Response · Authors · 2023-11-16
> **Reply to Reviewer ZH4m**
>
> We sincerely thank the reviewer for providing thoughtful review and positive feedback, and will update the paper accordingly. Below are our responses to the main questions by the reviewer.
>
> **1. How does 2D diffusion model jointly denoise multi-view images?**
>
> Multi-view images just stacked along the batch dimension and are fed into a ControlNet based model. Section A2.2 of new supplementary material gives more details of 2D denoising network.
>
> **2. How does post-optimization work?**
>
> The post-optimization is mainly based on SDS loss. We have supplemented more details of optimization phase in Section 3.5 of the revision paper.

---

### Official Review · Reviewer_hKcU · 2023-10-30

**Soundness:** 3 good
**Presentation:** 2 fair
**Contribution:** 3 good
**Rating:** 5
**Confidence:** 4

**Summary:**

The study introduces a diffusion model that simultaneously incorporates 2D and 3D object priors through a method termed "Bidirectional Diffusion." To enhance the synergy between the produced 2D multiview images and the 3D geometry, each denoising phase begins by consolidating the multiview images into a cost volume. These are then incorporated as conditions into the 3D diffusion segment, alongside the reference shape produced by ShapE. Conversely, the refined 3D geometry can be rendered back into images to steer the 2D multiview branch. By implementing this approach, the researchers illustrate the model's capability to efficiently produce 2D-3D congruent shapes in several diffusion steps, all within a minute. Tests conducted on the ShapeNet and Objaverse datasets reveal that the introduced method strikes an optimal balance between quality (as indicated by the CLIP score) and generation speed.

**Strengths:**

- The technique introduces a unique approach to integrating 2D image priors with 3D priors through a bidirectional diffusion model. Unlike conventional methods that primarily focus on distilling scores from image models, this technique actively harnesses the 3D prior, guiding it in tandem with the 2D diffusion process. This innovative approach sets it apart from other methods.

- This method adeptly draws upon priors from established foundational models, reducing the need for long-time additional training. Notably, the insights gained from ShapE and stable diffusion are seamlessly and effectively combined.

- Demonstrating speed and efficiency, the method can produce a commendable 3D shape based on textual guidance in just a matter of minutes.

**Weaknesses:**

- The paper's presentation lacks a clear focus, with undue emphasis on certain aspects that distract from its main contributions:
  1. While the authors highlight that their framework allows for the separation of shape and texture control, dedicating an entire page to this feature, it's evident that this separation isn't a core attribute of the proposed model. Instead, it's achieved through modifications in the text prompt and Shap-E guidance, diluting the impact of the claim.
  2. The authors showcase numerous impressive results achieved via 'optimization-based methods'. However, these results don't stem directly from their proposed pipeline but rather from the optimization methods. There's an insufficient description of these methods in the main content, and they fail to retain the fundamental shape or texture resulting from the bidirectional diffusion model.

- The strategy for enhancing priors is fundamentally a guidance system that doesn't rely on classifiers. This should be clarified by the authors.

- The quality of the generated results leaves room for improvement. The images appear blurry, making it challenging to discern the specific identity of the produced shape without cross-referencing the associated text prompt, as seen in the iKun example :) Additionally, the generated geometry exhibits numerous striped artifacts. This lack of clarity might stem from inadequate training and diminished resolution in the 3D domain. Employing a different foundational 3D representation might address these issues.

- The paper's writing quality needs enhancement. It contains several typographical errors, grammatical mistakes, and omitted cross-references. For instance, on Page 5, Fig. [] is missing, and on Page 6, the reference to "$N$ views" should be corrected to "$M$ views."

**Questions:**

- How is consistency between ShapE and the final outcomes (generated 3D shape) ensured?
- What's the rationale behind using SparseConvolution for the 3D diffusion branch? At the onset of diffusion, do you initiate with a dense 3D volume or a sparse one? If it's the latter, how is it initialized? Additionally, why in Eq.(5) is SparseConv applied to the function $\mathcal{G}$?
- On page 5, why does $\mathcal{V}$ have a prime superscript? In the context of bidirectional diffusion, what is the significance of the prime notation?

**Details Of Ethics Concerns:**

Not applicable.

---

> ### Author Response · Authors · 2023-11-16
> **Reply to Reviewer hKcU**
>
> We sincerely thank the reviewer for providing thoughtful review and positive feedback, and will update the paper accordingly. Below are our responses to the main questions by the reviewer.
>
> **1. Problem of separation of shape and texture control.**
>
> Please refer to Question 1 in interpretation of common questions.
>
> **2. Impressive results don't stem directly from their proposed pipeline.**
>
> Please refer to Question 3 in interpretation of common questions.
>
> **3. How is consistency between ShapE and the final outcomes (generated 3D shape) ensured?**
>
> Please refer to Question 2 in interpretation of common questions.
>
> **4. What's the rationale behind using SparseConvolution for the 3D diffusion?**
>
> The use of SparseConv follows SparseNeus. For a 3D cubic grid feature volume, some grid points may not project onto any rendering images. In addition, to reduce computation, some grid points corresponding to edge rays are selectively disabled. Therefore, using SparseConv in our 3D model is more flexible and efficient. The 3D diffusion process begins from sparse volume (grid points). Its initialization method is same as normal diffusion. We can regard the sparse volume as one-dimensional data. $\mathcal{G}\_{t\_0}$ represents a radiance field converted from the Shap-E latent code, specifically expressed as a 3D feature volume of size $s \times s \times s$, hence it can use SparseConv for feature mapping or further extraction.
>
> **5. Symbols.**
>
> $\mathcal{G}\_{t\_0}$ represents a radiance field converted from the Shap-E latent representations. It is expressed as a feature of a 3D grid of size $s \times s \times s$, hence it can use SparseConv for feature mapping or further extraction. $\mathcal{V}$ prime represents the sampling result of the network during the denoising process. We want to distinguish it from the ground truth represented by $\mathcal{V}$ without prime. Prime represents the result output by the network.

---

### Official Review · Reviewer_7ENe · 2023-11-02

**Soundness:** 3 good
**Presentation:** 1 poor
**Contribution:** 3 good
**Rating:** 6
**Confidence:** 4

**Summary:**

This method proposes a method that couples a 3D diffusion model and 2D diffusion model by cross-conditioning. The multi-view 2D diffusion model is conditioned on multi-view renderings from a denoised NeRF,  while the 3D diffusion model is conditioned on denoised multi-views from the 2D diffusion model. Through this coupling, the authors show that they can exploit the priors from both pretrained 3D diffusion models and 2D diffusion models.

**Strengths:**

1. Overall, the idea of coupling a 3D diffusion and a 2D diffusion model seems interesting and novel, as they can help each other. This is in contrast to the idea of render-diffusion work where a image-to-3D reconstructor is used as a denoiser.
2. It’s also interesting to see the use of pretrained 3D diffusion models (like Shap-E) and 2D diffusion models (like DeepFloyd) to exploit their priors.
3. Separate control of shape and texture generation seems very cool!

**Weaknesses:**

1. The paper seems to be done in a rush with many typos.
     - “To conquer the challenges…” paragraph of the introduction, duplicate “ensures”
     - Top of page 5: figure is not referenced correctly
     - Sentence after Eq. 1, noise scale should be monotonically increasing?
     - Top of page 6, N is abused for denoting both grid resolution and number of views. In Eq. 6, M is used to denote # views.
     - Beginning of section 3.3: “To 2D-3D consistency” —> “To encourage 2D-3D consistency”?

 2. Many details are missing
    - Model architecture for the 3D denoiser? Whole model is trained for how long with what learning rate? Batch size?
    - How is Objaverse 40k subset selected out of the total 780k objects? What are the data curation criteria?
    - In Eq. (5), G_{t_0} seems to be an MLP according to G(p)=MLP(\lambda(p); \theta=C). How is Sparse3DConv applied to a MLP? In addition, what is the t_0 set to?
    - The ablation talks about fixing 3D prior or 2D prior. How is this fixing done? Does it mean that non-stochastic DDIM inference is used and the initial noise is fixed?
    - Section 4.2 talks about 50 prompts for evaluation. How are these 50 prompts designed?
    - The training requires paired (captions, multi-view renderings) data? If so, how are the captions generated?
    - How is the Shap-E latent obtained during training? By feeding one image to Shap-E or the text prompt to Shap-E? If it’s the latter one, how to address the inconsistency between Shap-E generated shapes and the multi-view images?

**Questions:**

Please see my comments in the weakness section.

---

> ### Author Response · Authors · 2023-11-16
> **Reply to Reviewer 7ENe**
>
> We sincerely thank the reviewer for providing thoughtful review and positive feedback, and will update the paper accordingly. Below are our responses to the main questions by the reviewer.
>
> **1. How is this fixing done?**
>
> Please refer to Question 1 in interpretation of common questions.
>
> **2. How is the Shap-E latent obtained during training?**
>
> Please refer to Question 2 in interpretation of common questions.
>
> **3. Issue of Eq. (5).**
>
> In Eq. (5), $ \mathcal{G}\_{t\_{0}} $ represents a radiance field converted from the Shap-E latent code with $t\_0$ step noise. To avoid confusion, we have clarified definition of $\mathcal{G}$ to: We design a feature volume $\mathcal{G}$ to represent a radiance field converted from the latent code $\mathcal{C}$. And Section 3.2 of main paper explains the meaning of $t\_0$: To generate 3D objects beyond Shap-E model, we add Gaussian noise at level $t_0$ to the clean latent code, resulting in the noisy latent representation $\mathcal{C}\_{t\_0}$, where $t\_0$ represents a predefined constant timestep. Subsequently, the noisy radiance field $\mathcal{G}\_{t\_0}$ is decoded by substituting $\mathcal{C}$ with $\mathcal{C}\_{t\_0}$. This design establishes a coarse-to-fine relationship between the 3D prior and the ground truth, prompting the 3D diffusion process to leverage the 3D prior without becoming excessively dependent on it.

---

### Author Response · Authors · 2023-11-16
**Interpretation of some common questions in comments**

**1. Revision paper.**

We have updated a revision paper and supplementary material to fix writing problems and add more details of our work.

**2. Problem of separating shape and texture control. How to fix the prior?**

We have supplemented the details of separating texture and geometry control in the revised version of Section 3.4. Specifically, to seperate this control, on the one hand, we first fix both the initial 3D noisy SDF grids and the conditional radiance field $\mathcal{C}_{t_0}$ while enlarging its influence by 3D prior enhancement. In this way, we can modify the 2D diffusion process by adjusting the text prompts to change texture while maintaining overall shapes; on the other hand, we can keep the texture styles by maintaining keywords in text prompts and enlarge its influence by 2D prior enhancement, then the shape can be adjusted by modifying the 3D diffusion process like varying conditional radiance field.

**3. How to ensure the consistency between Shap-E and the final outcomes?**

During training, To obtain the latent code $\mathcal{C}$ for each object, we use the encoder in Shap-E to encode each object and apply $t_0=0.4$ level Gaussian noise to $\mathcal{C}$ to get noisy $\mathcal{C}_{t_0}$, and then decode the condition radiance field during training. Therefore, it does not have a huge discrepancy from the training target. We have supplemented more details in A.1 and A2.1 of supplementary material.

**4. Explain the gap between the optimized results and the results before optimization, and details of optimization.**

In fact, BiDiff is a framework oriented towards practical application. Undeniably, optimization-based methods surpass feed-forward methods in result quality, but this comes at the cost of lengthy optimization times, normally taking several hours. Our motivation is that lengthy optimizations, taking several hours, prevent creators from extensively adjusting prompts to obtain optimal results, as they would with 2D generative models. To address this, BiDiff enables creators to rapidly adjust prompts to obtain a satisfactory preliminary 3D model preview, taking only 40 seconds for each trial, through a lightweight feed-forward generation process. Although it is lightweight, it still ensures excellent conditional control and can directly yield 3D results, not just multi-view images. These capabilities surpass those of previous feed-forward models. Once the optimal preview result is found, users can further refine it into high-fidelity results using optimization method.

Moreover, with the initial geometry provided by BiDiff, optimization methods consume less time to obtain high-quality 3D results, and also avoid common issues like multi-face when running them from scratch. This design contributes to the real-world application of 3D generation technology by exploring the feasibility of this approach, combining the advantages of both technological pathways, similar to image generation. We have supplemented more details of optimization phase in Section 3.5 of revision paper.